# Statistical Evidence for a Helical Nascent Chain

**DOI:** 10.3390/biom11030357

**Published:** 2021-02-26

**Authors:** Leonor Cruzeiro, Andrew C. Gill, J. Chris Eilbeck

**Affiliations:** 1CCMAR/CIMAR—Centro de Ciências do Mar, FCT, Universidade do Algarve, Campus de Gambelas, 8005-139 Faro, Portugal; 2School of Chemistry, Joseph Banks Laboratories, University of Lincoln, Green Lane, Lincoln LN67DL, UK; AnGill@lincoln.ac.uk; 3Department of Mathematics and Maxwell Institute, Heriot-Watt University, Edinburgh EH14 4AS, UK; J.C.Eilbeck@hw.ac.uk

**Keywords:** protein folding, single amino acid distributions, secondary structure prediction, folding pathway

## Abstract

We investigate the hypothesis that protein folding is a kinetic, non-equilibrium process, in which the structure of the nascent chain is crucial. We compare actual amino acid frequencies in loops, α-helices and β-sheets with the frequencies that would arise in the absence of any amino acid bias for those secondary structures. The novel analysis suggests that while specific amino acids exist to drive the formation of loops and sheets, none stand out as drivers for α-helices. This favours the idea that the α-helix is the initial structure of most proteins before the folding process begins.

## 1. Introduction

The protein folding problem consists of trying to obtain the three dimensional native structures of proteins from their amino acid sequences. This can be pursued in essentially two ways. One way is to devise a set of rules or an algorithm to obtain the native structure from the amino acid sequence, and a second way is to determine the physical forces that take the nascent chain to the native state. The first way has been pursued since 1974 [1] and has recently lead to very remarkable protein structure predictions (see predictioncenter.org (accessed on 15 December 2020) and especially the results from CASP14) and to claims that the protein folding problem is solved. However, even very sophisticated black box approaches cannot enlighten us about the physical forces that drive protein folding. On the other hand, such forces, once identified, constitute the complete answer and should allow us to predict native structures as well. The ultimate aim of the work in this present paper is to understand the physical process of protein folding.

Since the thermodynamic hypothesis was first proposed [2,3], the guiding idea behind most protein folding studies has been that the native state is uniquely specified by the amino acid sequence. More than five decades of studies of protein re-folding have lead to the idea that proteins can fold to their native state, spontaneously, from any initial structure, including fully extended and disordered conformations, via a process of free energy minimization (see, e.g., [4,5,6,7,8,9,10,11]). On the other hand, a growing body of evidence from studies of protein folding in the cell shows that nascent chains acquire structure while still inside the ribosome [12,13,14,15,16,17,18,19,20]. Yet, most algorithms for secondary structure prediction continue to apply the thermodynamic hypothesis according to which some amino acids (or, in a finer analysis, some amino acid sequences) do, for some reason, lead to the formation of helices, while others lead to the formation of sheets and loops. Accordingly, the aim of statistical analyses of the correlations between sequence and structure has been to find such structure-defining amino acids (or amino acid sequences).

One difficulty with the quest of getting three dimensional structures from sequences is the variety of sequences that lead to very similar structures. Indeed two proteins with only 30% sequence similarity have a strong probability of sharing very similar three dimensional structures [21]. Thus, instead of just a few amino acid patterns we can have many amino acid patterns that, in cells, lead to the same structural result (and the reverse can also happen, similar sequences can lead to different structures [22]). Another difficulty is that protein native structure may be one of the many kinetic traps into which the same polypeptide can find itself in, as shown in [23,24,25] and proposed in [26,27,28,29,30]. In this case, reproducibility in always reaching the same native structure can be achieved if the initial structure and the pathway followed from it are always the same, as explained in detail in [28]. Thus, the purpose of our statistical analyses is to infer the structure of the nascent chain and of the generic features of the pathway.

In Section 3, we probe the sequence-structure variety by calculating the distributions of amino acid frequencies in the three main secondary structures over a population of 13,413 proteins. Furthermore, in order to extract the real bias that each amino acid may have for or against a given secondary structure, we compare the existing protein secondary structures to ideally unbiased ones. Although this kind of analysis was already made in the pioneering statistical analysis of protein structures by Chou and Fasman [1], here we revisit it in a different spirit. Indeed, whereas usual structure-sequence analyses aim at determining the final native structure, here, guided by the kinetic hypothesis, we use them to try to determine the initial structure, that is, the structure of the nascent chain. While the experimental evidence suggests that the nascent chain can be either α-helical [12,13,14,15,16,17,20] or a more extended conformation [12,17,18,19], we propose that the simplest interpretation of the results we obtain is that the nascent chain of most proteins is α-helical. We also propose that a more fruitful way of solving the protein folding problem is to determine the pathway that proteins follow in going from the initial helix to the native state. To that end, a generic pathway that can be inferred from our results, is also presented.

## 2. Materials and Methods

An ss.txt file with the sequences of 444,520 proteins with known structure was obtained from the protein data bank [31]. For each protein included in the ss.txt file we have its sequence followed by the corresponding secondary structure type of each amino acid, as assigned by the DSSP (Define Secondary Structure of Proteins) program [32,33]. In order to reduce redundancy, a list of 14346 proteins with less than 25% sequence identity (file cullpdb_pc25_res3.0_R1.0_d190321_chains14346) was obtained from the PISCES site [34] and the proteins common to both of the two files were selected. This led to the 13,413 proteins listed in the file list-of-13413-proteins.dat. This is the set of proteins used in the analyses described in Section 3. In the file Appendix A show the results that are obtained when the set of all the 444,520 proteins, listed in the file list-of-444520-proteins.dat, is used.

## 3. Results

We start by determining the average amino acid composition of the proteins in our protein set. Let $n_{asp}$ be the number of amino acids *a* found in secondary structures *s*, in a given protein *p*, (here p=1,⋯,M, with *M* = 13,413 being the total number of proteins in the set). This set includes only proteins with the twenty most common amino acids so that a= (A (Alanine), C (Cysteine), D (Aspartic Acid), E (Glutamic Acid), F (Phenylalanine), G (Glycine), H (Histidine), I (Isoleucine), K (Lysine), L (Leucine), M (Methionine), N (Asparagine), P (Proline), Q (Glutamine), R (Arginine), S (Serine), T (Threonine), V (Valine), W (Tryptophan), Y (Tyrosine)). From the number, $n_{asp}$, we get the sequence size of protein *p*, $n_{p}$, by:(1)$$n_{p}=\sum_{a,s}n_{asp},$$
and the number, $n_{a}\left(p\right)$, of amino acids *a* in protein *p*:(2)$$n_{a}\left(p\right)=\sum_{s}n_{asp}.$$

The average abundance of each amino acid in protein set, ${\bar{f}}_{a}$, can be calculated as:(3)$${\bar{f}}_{a}\left(a\right)=\frac{1}{M}\sum_{p}\frac{n_{a}\left(p\right)}{n_{p}}$$

In many studies (see, e.g., [1,35,36]), all proteins are mixed together into one single enormous protein and the statistics are calculated for this single protein. However, such a single protein does not exist and it is impossible to know what its structure would be. This is one reason why, in Equation (Equation 3) as well as in all averages over the protein set that are mentioned below, we consider the statistics for each protein separately, and then average over all the proteins in the set. A second reason is that, as found in [36] and as illustrated in Figure 2 below, those separate statistics can vary significantly from protein to protein.

Figure 1 displays the average abundance ${\bar{f}}_{a}$ of each amino acid in our protein set as percentages so that summing them over all amino acids leads to 100.

It shows that some amino acids appear more abundantly than others. Thus, W, C, M and H appear less frequently, while L, A, E, G, V, S, D and T are more frequent, as is usually found [36]. However, it is also known that the average amino acid abundance depends on protein size [36]. The protein set used here includes proteins with sizes from 20 to 1859 amino acids, with a broad peak at 157. Comparing with values obtained in [36] for proteins with an average size of 200 amino acids, the abundances are similar. Furthermore, using the larger data set mentioned in Section 2, the results are virtually indistinguishable (compare Figure 1 above with Appendix A). This validates our protein set from the point of view of average amino acid composition.

In the absence of any bias, the abundances of the amino acids in each secondary structure should be very similar to those displayed in Figure 1 (and they would be exactly equal if all proteins had the same amino acid composition and the same percentages of secondary structures). Thus, a first measure of the bias of an amino acid for a particular secondary structure can be obtained by comparing the average abundance of that amino acid, as shown in Figure 1, with the correspondent abundance in that secondary structure.

To that end we calculate the frequency, f(a,s,p), of finding amino acid *a* in secondary structure *s* in protein *p* as:(4)$$f\left(a,s,p\right)=\frac{n_{asp}}{n_{p}}.$$

The DSSP program [32,33] considers eight different types of secondary structures, namely, H (α-helix), E (β-sheet), G (3/10 helix), I (π-helix), B (β-bridge), T (turn), S (bend) and C or space (random coil). Here we concentrate on the most ubiquitous secondary structures and *s* will comprise just four types, namely *s* = H, E, L, (G+I), where L stands for loops (which include the secondary structure types B, T, S, C or space in DSSP [32,33]). In this way, we separate the helices and sheets from the less structured regions that connect them. Furthermore, our calculations showed that helices G plus I contribute only up to 3% of the total in each protein and thus results for them are omitted in the figures below.

From the frequencies, f(a,s,p) in Equation (Equation 4), we can determine the average abundance of each amino acid *a* in each secondary structure *s*, $\bar{f}\left(a,s\right)$, by making the average over the protein set:(5)$$\bar{f}\left(a,s\right)=\frac{1}{M}\sum_{p}f\left(a,s,p\right).$$

However, as the frequencies f(a,s,p) vary considerably from protein to protein, their average values, $\bar{f}\left(a,s\right)$, just by themselves, are a poor representation of their full distribution. To demonstrate the variety of values that the frequencies f(a,s,p) can assume in the proteins of our set, a few selected distributions are displayed in Figure 2, where red is for s=α-helices, blue is for β-sheets and green is for loops, and the amino acid *a* selected is specified at the top of each plot.

This figure shows, for example, that in the case of a= P (Pro) and s=α-helix or β-sheet, the maximum of the distribution is at f(a,s,p)=2. The maximum corresponds to the most probable event and the number two means that the most probable event is that of proteins with sequences in which 2% of the amino acids are P’s located in helices or in sheets. On the other hand, we expect to find more P’s in loops and Figure 2 does indeed show that, in the case of a= P and s= loops, the maximum is at 6, which means that the most probable event is that of proteins in which 6% of the all their amino acids are P’s located in loops.

Figure 2 also shows that, for each of the amino acids *a* and of the secondary structures *s*, the frequencies f(a,s,p) can have different values in different proteins, i.e., the fact that the ordinate for f(a,s,p)=6 is not zero for a= P and s=β-sheets means that there is a small number of proteins in which 6% of the all their amino acids are P’s located in β-sheets (and the same for α-helices). This variety can be measured by the full width at half maximum (FWHM) of each distribution. The FWHM is determined by going down from the maximum, in each direction, until we reach a value of y-coordinate (i.e., a value of the distribution) that is half the value of the maximum. The FWHM is a measure of the uncertainty around the most probable value and in Figure 2 it is marked by the horizontal dotted lines. For example, in the case of a= P and s= loops, the FWHM is 7.4, and we can calculate that the “area” under the green curve, comprehended between the lower and the higher extremities of that dotted line, is 77.6, which means that, for 77.6% of the proteins in the set, between 3.3% and 10.7% of their amino acids are P’s in loops. Therefore, although the most probable event is constituted by proteins in which 6% of the all their amino acids are P’s located in loops, there is a non-negligible number of proteins for which this percentage can be as low as 3.3% or as high as 10.7%. The broader a distribution (the greater FWHM), the greater the variety of values of the frequencies f(a,s,p) found in the different proteins.

Furthermore, Figure 2 shows that the distributions for f(a,s,p) are skewed, with longer tails towards the larger numbers, for all amino acids in all secondary structures. Thus, in Figure 3, instead of the first moment of the distribution (the average defined by Equation (Equation 5)), we plot the most probable value (the position of maximum of the corresponding distribution, signalled by a marker), and instead of the square root of the second moment (the standard deviation), we plot the FWHM to quantify the uncertainty around the most probable value. Note that while we have found that the average values $\bar{f}\left(a,s\right)$ (not shown, see Equation (Equation 5)) are very similar to the most probable values displayed in Figure 3, so that the positions of the markers can also be interpreted as average values, the standard deviations are different from the FWHM, with the latter providing a more accurate representation of the uncertainty above and below the average. Indeed, while the standard deviations are more meaningful for symmetric distributions and would lead to equal intervals above and below the average, the FWHM reproduces the skewness of the distributions, with values larger than the average being more probable than values below, as was already apparent in Figure 2.

Inspection of Figure 3 shows that none of the three curves in it is similar to that in Figure 1, which means that the amino acid distributions in each secondary structure are biased, as expected [1,35]. The most similar is arguably that for α-helices, which, if it had higher values for G and P, and a lower value for R, would have a shape close to the curve in Figure 1. On the other hand, with the exception of the low abundant amino acids W, C, M and H, the absolute values, even for the α-helix curve, are different. Indeed, the average values in Figure 3 suggest that α-helices are characterized by larger amounts of L, A and E, while β-sheets are characterized by larger amounts of V, L, and I, and loops have more G, S, and D. It is tempting to equate a greater number of amino acid *a* in a given secondary structure *s* with a propensity for that *a* to induce the formation of *s*. However, variables like the average frequency $\bar{f}\left(a,s\right)$ can be inappropriate for at least two reasons. One reason is that the average abundance, ${\bar{f}}_{a}$, (cf. Equation (Equation 3)) is not the same for all amino acids, as shown in Figure 1. A second reason which will skew average amino acid frequencies is that the three secondary structures do not appear in the same amounts in every protein. The average abundance of the each secondary structure *s* in the set, ${\bar{f}}_{n}\left(s\right)$, can be determined by:(6)$${\bar{f}}_{n}\left(s\right)=\frac{1}{M}\sum_{p}\frac{n_{s}\left(p\right)}{n_{p}}$$
with $n_{s}\left(p\right)$, the number of sites with secondary structure *s* in protein *p*, being
(7)$$n_{s}\left(p\right)=\sum_{a}n_{asp}.$$

We calculate that in our protein set loops are the most frequent secondary structures (44% on average), followed by α-helices (33%), which in turn are followed by β-sheets (20%). Again, all things being equal, these different percentages will tend to lead to greater probabilities for all amino acids to appear in loops. However, even more important than those two reasons for the skewing of the abundance of each amino acid in the different secondary structures is the fact that, when measuring the bias of one amino acid for a specific secondary structure, the control should be what would happen in the complete absence of that bias. Here, this is done by comparing the actual number of amino acids *a* in secondary structures *s* in protein *p*, $n_{asp}$, with the number, $E_{asp}$, that would be expected to arise if the same amount of amino acid *a* and the same amount of secondary structure *s* were distributed in a completely random fashion in that protein. i.e., the bias of an amino acid *a* to a secondary structure *s* in a protein *p* is estimated by the ratio, R(a,s,p), given by:(8)$$R\left(a,s,p\right)=\frac{n_{asp}}{E_{asp}}$$

This estimate involves not only a proper control for the bias but has also the advantage of eliminating the skewness in the different abundances of amino acids or secondary structures because, for each protein *p*, these abundances appear in equal measure in the numerator and denominator of the ratios R(a,s,p) (see Equation (Equation 8) and the equations below).

With this definition, a ratio R(a,s,p) of approximately 1 for a given protein means that the distribution of amino acid *a* among the secondary structure *s* is approximately random, that is, unbiased. A ratio greater than 1, on the other hand, means that that amino acid appears more often than would be expected and therefore has a positive bias for the secondary structure *s*, and a ratio lower than 1 means that that amino acid appears less often than would be expected and therefore has a negative bias for that secondary structure *s*.

Let us then calculate the ratio R(a,s,p) (cf. Equation (Equation 8)). Designating the random uniform (unbiased) distribution for finding amino acids *a* in secondary structure sites *s* by r(a,s,p), the estimated number, $E_{asp}$, of *a* in *s*, in the absence of bias, is:(9)$$E_{asp}=n_{p}\;r\left(a,s,p\right)$$

Substituting Equation (Equation 9) in Equation (Equation 8) we get:(10)$$R\left(a,s,p\right)=\frac{f(a,s,p)}{r(a,s,p)}$$

In the absence of any correlation between amino acids and secondary structures, that is, in the absence of any bias, the probability, r(a,s,p), of finding amino acid *a* in secondary structure *s* in a given protein *p*, is the product of the probability, $r_{a}$, of finding that amino acid in any site of the protein, with the probability, $r_{s}$, of finding structure *s* in any site of the protein:(11)$$r\left(a,s,p\right)=r_{a}\;r_{s}$$

In an unbiased distribution, all amino acids *a* have equal probability of appearing everywhere and all secondary structure sites *s* have also equal probability of appearing everywhere. Thus, $r_{a}$ is the number, $n_{a}\left(p\right)$, of amino acids *a* in protein *p* (see Equation (Equation 2)), divided by the total number of sites in the protein:(12)$$r_{a}=\frac{n_{a}\left(p\right)}{n_{p}}$$
and $r_{s}$ is the number, $n_{s}\left(p\right)$, of secondary structure *s* sites in protein *p* (see Equation (Equation 7)), divided by the total number of sites in the protein:(13)$$r_{s}=\frac{n_{s}\left(p\right)}{n_{p}}.$$

Substituting Equations (Equation 12) and (Equation 13) in Equation (Equation 11), the random probability r(a,s,p) becomes:(14)$$r\left(a,s,p\right)=\frac{n_{a}\left(p\right)}{n_{p}}\;\frac{n_{s}\left(p\right)}{n_{p}}.$$

Using Equations (Equation 2), (Equation 7) and (Equation 14) it is easy to show that $\sum_{a,s}r\left(a,s,p\right)=1$, as must be for r(a,s,p) to be a probability.

Equations (Equation 10), together with Equations (Equation 14) and (Equation 4), allows us to determine the ratios R(a,s,p) for each protein, except when a protein lacks amino acid *a* (in which case $n_{a}\left(p\right)=0$) and/or secondary structure *s*, (in which case $n_{s}\left(p\right)=0$), leading to that both f(a,s,p) and r(a,s,p) are equal to zero. Then, the ratio R(a,s,p) (Equation (Equation 10)) is undetermined and is thus not included in the calculations.

As happens for the frequencies f(a,s,p), also the ratios R(a,s,p) (Equation (Equation 10)) can vary much from protein to protein. Figure 4 displays a few of the distributions of the ratios. The vertical dotted lines mark the R(a,s,p)=1 values, which, as explained above, indicate an absence of bias of *a* towards *s* in the corresponding proteins. On the other hand, when R(a,s,p)>1, i.e., for proteins that contribute to the points above the vertical line, amino acid *a* appears in the secondary structure *s* in greater numbers than would be predicted in the absence of bias and, from those proteins, we would conclude that *a* is structure-forming for that secondary structure *s*. Similarly, when R(a,s,p)<1, i.e., for proteins that contribute to the points below the vertical line, amino acid *a* appears in secondary structure *s* in smaller numbers than would be predicted in the absence of bias and, from those proteins, we would conclude that *a* is structure-breaking for that secondary structure *s*. Figure 4 shows that these assignments are fuzzy because for any given amino acid *a* and any given secondary structure *s* we can find proteins in which *a* is structure-neutral for *s*, namely, those for which R(a,s,p)=1, as well as other proteins in which the same *a* is structure-forming for the same *s* and yet other proteins in which it is structure-breaking. For instance, according to the definition above, for a= Ala and s=α-helix, in the proteins that contribute to the part of the histogram to the right of the vertical line, Ala is structure-forming, in the proteins that contribute to the point where the vertical line intersects the histogram Ala is neutral, and in the proteins that contribute to the part of the histogram to the left of the vertical line, Ala is structure-breaking. A more balanced definition, that reflects better the variety of behaviour in the different proteins, is to consider structure-forming the amino acids for which the greater part of the corresponding histogram lies above the vertical line (as happens for a= Ala, Glu, Leu in s=α-helix, and for a= ILE, Leu and Val in s=β-sheet and for a= Asp, Gly, Pro and Ser in loops, considering the histograms in Figure 4, as well as those in Appendix A). Furthermore, we should also distinguish between strong structure-formers, like Ile and Val in β-sheets, in which not only the most probable value of R(a,s,p) is clearly above 1, but also all points along the FWHM are above that value and cases like Ala and Leu, which, although being structure-formers for α-helices, are more weakly so.

One difference between the distributions in Figure 2 and those in Figure 4 is that the latter are bimodal, with an extra peak at R(a,s,p)=0. R(a,s,p)=0 means that the corresponding protein possesses amino acid *a* and also possesses secondary structure *s*, but it does not possess amino acid *a* in secondary structure *s*, in spite of a non-zero random probability for that to happen (see Equation (Equation 14))). Although only a few distributions are displayed in Figure 4, we have verified that such peaks at zero are present in all 60 distributions that can be obtained for the 20 amino acids in s=α-helix, β-sheet and loop. In many cases, as for P (Pro) in α-helices and for A (Ala) and P (Pro) in β-sheets, the peak at zero is the mode of the distribution, i.e., R(a,s,p)=0 for those amino acids in those secondary structures is the most probable event. In these cases, the FWHM is effectively zero.

In Figure 5, we apply the same criteria as before, and plot the most probable values of the ratios R(a,s,p) (see Equation (Equation 10)), and take the FWHM as an estimate of the uncertainty around those values.

The cases in which the distribution of the ratios R(a,s,p) has a peak at zero that is higher than the one in the middle are very clearly identifiable in Figure 5: they are those for which the most probable value is zero. The secondary structure with the greater number of amino acids in that category is β-sheets, with 17 such amino acids, followed by α-helices with 15 such amino acids, followed by loops with only such seven amino acids. The amino acids and secondary structures with non-zero values are those for which the peak in the middle is higher than the one at zero.

In a bimodal distribution, the criterion of using the most probable value may overemphasise values that are not very frequent, if the “area” under the respective peak is small when compared with the area under the other peak. Although in the case a= Pro and s=α-helix, the peak at zero has a height 0.32, meaning that R(a,s,p)=0 in 32% of the proteins in the set, and for s=β-sheet it is 40%, for the majority of the amino acids *a* and secondary structures *s*, the peak at zero accounts for less than 20% of the proteins in the set. Furthermore, it is noticeable that the importance of the peaks at zero is greater for the secondary structures that are less common, being most pronounced for β-sheets and decreasing for α-helices and loops. This indicates that in a larger set these peaks may decrease in size. Thus, in Figure 6 the most probable values and the FWHM have been calculated using only the middle peaks in the histograms, which, for most *a* and *s*, represent 80% or more of the proteins in the set.

We have already explained above how we can identify structure-forming amino acids, and distinguish between strong and weak structure-formers. In Figure 6, a strong structure former is an amino acid whose most probable value of the ratio R(a,s,p) ( Equation (Equation 10)) is clearly above 1 and for which the uncertainty (FWHM) is also clearly above 1. This happens for V (Val) and I (Ile) in β-sheets, as has already been pointed out before, and also for G (Gly) and P (Pro) in loops. Similarly, a strong structure-breaker is an amino acid whose most probable value of the ratio R(a,s,p) ( Equation (Equation 10)) is clearly below 1 and for which the uncertainty is also clearly below 1. In Figure 6, for s=α-helix we identify two such amino acids namely, G (Gly) and P (Pro), for s=β-sheet P (Pro) and D (Asp), and more tangentially, G (Gly), and for s= loops, also tangentially, I (Ile), V (Val) and L (Leu).

Figure 6 also shows that the majority of the amino acids have most probable values either above or below the R(a,s,p)=1 line, together with uncertainties that cross that line. In cases like A (Ala) and L (Leu) and less obviously so for E (Glu) in α-helices, in which not only the most probable value but also the greater part of the FWHM are above 1, such amino acids should be considered structure-formers, albeit weak ones. Furthermore, similarly, in cases like N (Asn) and E (Glu) in β-helices, in which not only the most probable value but also the greater part of the FWHM are below 1, such amino acids should be considered weak structure-breakers. On the other hand, when the FWHM is approximately equally spread above and below the R(a,s,p)=1 line, and the most probable value is close to it, either above or below, such amino acids should more properly be considered neutral with respect to the formation of the corresponding secondary structure *s*. Inspection of Figure 6 shows that the majority of the amino acids are of the latter type, with the α-helix possessing 12 neutral amino acids, and loops and β-sheets possessing 10 each.

Keeping the previous definitions in mind when comparing Figure 6 with Figure 3 we notice several differences. First, L, which according to Figure 3 might be considered a strong structure-forming amino acid for both α-helices and β-sheets and neutral for loops, is revealed as only weakly structure-forming for α-helices, neutral for β-sheets and structure-breaking for loops. Similarly, A, which according to Figure 3 might be considered a strong structure-forming amino acid for α-helices is revealed as only weakly structure-forming. Finally, E, which according to Figure 3 might be considered a strong-structure former for α-helices is revealed as a neutral one.

Inspection of Figure 6 shows that β-sheets possess two strong structure-formers, namely, V and I, three weak structure-formers, namely, F, C and W, three strong structure-breakers, P, D and G, and two weak structure-breakers, N and E. Loops have two strong structure-formers, namely, G and P, four weak structure-formers, namely, N, D, and S and, to a lesser extent, H, three strong structure-breakers, I, V and L and one weak structure-breaker, F. Furthermore, α-helices have three weak structure-formers, namely, A, L and to lesser extent, E, two clearly strong structure-breakers, namely, G and P, and three weak structure-breakers, namely, N, D and S. From this point of view, α-helices stand out as the secondary structure with the least number of structure-forming amino acids. Indeed, while loops have six structure-forming amino acids, two of which are strong, and β-sheets have five, two of which are strong, α-helices have three structure-formers, all of which are weak. How is it that the second most ubiquitous secondary structure does not possess strong structure-forming amino acids? It may be argued that it is because single amino acids are not sufficient to determine the secondary structure and that amino acid sequences must be considered. This is certainly true, but this applies equally to β-sheets and loops. We expect that a secondary structure in which we can identify single amino acids as important for structure formation should arise more readily than a secondary structure in which this does not happen. This expectation is fulfilled in the case of loops which are the most populated secondary structures in proteins and are also those which possess the greater number of structure-forming amino acids. However, the fact is also that, in spite of its greater number of structure-forming amino acids, β-sheets are less populated than α-helices, with which they share three of their structure-breakers (P, D and G). In the next section, we propose one explanation for these findings.

## 4. Discussion

For more than four decades, the experimental knowledge about protein folding came from experiments in which re-folding (or its absence) is followed after the action of chaotropes. To many researchers these experiments have suggested that the initial structure is immaterial and that proteins are able to return to their native state even from completely unstructured states [2,3,4,5,6,7,8,9,10,11]. In contrast, more recently, folding studies have suggested that the nascent chain is structured [12,13,14,15,16,17,18,19,20], and that, in many cases, it is α-helical [12,13,14,15,16,17,20]. It is also curious that the membrane regions of membrane proteins tend overwhelmingly to be made of helices. Since they are protected as soon as they emerge from the ribosome and until they are inserted in the membrane, and since they denature when taken out of the membrane, it does seem that they are synthesized in the shape of helices to start with. Furthermore, a specific mechanism for the formation of α-helices has been demonstrated for 5 different polypeptides in molecular dynamics simulations [30], namely, a forced rotation on the C-terminal, originating within the ribosome, lead to the formation of α-helices when the N-terminal outside is restrained [30]. Although the folding conditions are different and, the system undergoing folding, the protein, is the same, and the physical principles that rule its equilibrium dynamics and stability must be the same in both cases. Thus, understanding how folding takes place in the cell must necessarily be relevant to all the other forms of folding.

In cells, all proteins, whatever their sequence, are synthesized by the same machine, the ribosome, and it is very probable that this machine follows the same mechanism for all sequences so that all nascent chains start with the same structural constraints. Only two secondary structures fit within the ribosomal exit tunnel: linear or helical. Let us consider each of these possibilities separately.

First, let us consider that all proteins are synthesized as linear, unstructured, polymers and that α-helices and β-sheets form later. To evolve from such initial long loops to other secondary structures, proteins would need strong structure-breakers for loops that would be at the same time strong structure-formers, some for α-helices, and others for β-sheets. Furthermore, since α-helices would be competing with β-sheets, the structure-formers of the former should also be structure-breakers of the latter, and vice versa. Figure 6 shows that two of the strong structure-breakers for loops, V and I, are also strong structure-formers for β-sheets, and that the one weak breaker for loops, F, is a weak former for β-sheets. Thus, if the initial structure were disordered, we would predict that regions with extra amounts of V and/or I and/or F would have a reasonable probability of turning into β-sheets.

What about α-helices? Loops have another strong structure-breaker, namely, L, but this amino acid is only a weak structure-former for α-helices. Moreover, the other (weak) formers for α-helices, namely, A and E, are structure-neutral for loops. Thus, regions rich in A and/or L and/or E might evolve into α-helices but they might just as probably remain disordered. Furthermore, because four of the structure-breakers of β-sheets (P,D,G and N) are also structure-breakers of α-helices, it is very unlikely that β-sheets would evolve into α-helices. Therefore, if the initial structure were disordered it would be difficult to understand why α-helices are more prevalent than β-sheets. In fact, we should expect protein structure to be essentially disordered, interspersed with β-sheets in regions with greater amounts of V and I, and with the occasional loose α-helix in regions with substantial amounts of A and/or L and/or E.

Let us now consider the alternative case in which the nascent chain is α-helical. In this case, to evolve into loops and β-sheets, proteins would need strong structure-breakers for α-helices that would be at the same time strong structure-formers, some for loops, and others for β-sheets. Furthermore, since loops would be competing with β-sheets, the structure-formers of the former should also be structure-breakers of the latter, and vice versa. Figure 6 shows that the two (strong) structure-breakers for α-helices, G and P, are also strong structure-formers for loops, and that the three weak breakers for α-helices, N, D and S, are also weak formers for loops. Thus, if the initial structure were α-helical, we would predict that regions with sufficient amounts of G and/or P and/or N and/or D and/or S would have a very reasonable probability of turning into loops.

On the other hand, because strong and weak structure-breakers of α-helices are also structure-breakers of β-sheets in equal amount, it is very unlikely that the regions of the initial α-helix that are rich in helix-breakers would evolve directly into β-sheets. However, regions rich in D and/or N and/or S might evolve first from α-helices into loops and, if that region were also rich in V and/or I, then from loops into β-sheets. However, because D, N and S are only weak α-helix breakers and weak loop-formers, and a double condition must be verified, namely, to create the loop intermediate first and later the β-sheet, we would expect such transformations not too occur very often.

From the two previous paragraphs we conclude that, if the nascent chain is α-helical we can explain that loops are still the most prevalent secondary structures from the fact that the five helical structure breakers are all loop formers. We can also explain why β-sheets are the least prevalent because they would not evolve directly from the helix and would require the loop as an intermediate, and would only evolve from that intermediate in regions with sufficient amounts of V and/or I. Furthermore, finally, we would also explain why the helix, in spite of its lack of strong formers, is still the second most abundant secondary structure. Indeed, if the α-helix is there from the start, it needs only to be stable enough to survive. In this case, having 12 structure-neutral amino acids, which would be a negative factor when a structure needs to be formed, becomes a positive factor when the structure is already there. The proposal in [27] is that the helix forms while still inside the ribosome but another possibility is that it forms right outside as a result of constraining the nascent chain while rotating it inside the ribosome [30].

When we consider only the single amino acid distributions, as is done in this study, we conclude that the possibility that fits better with the results obtained is that the protein nascent chain is helical. Indeed, some recent discussions of ribosome evolution suggest that the exit tunnel has evolved to favour formation of helical segments [37]. The dimensions of this tunnel place restrictions on the secondary structural elements that can form in nascent chains during translation [38], particularly in the first 50 Å or so from the peptidyl transfer complex where the diameter is tightly constrained [39,40,41]. In recent years a range of sophisticated biochemical and biophysical tools have been developed to study the structure of nascent chains complexed inside the exit tunnel, largely driven by the availability of cell-free translation preparations and by our ability to pause translation in controlled manners. Analytical tools that have been applied include cryo-electron microscopy, protein labelling and crosslinking approaches, nuclear nagnetic resonance (NMR) and mass spectrometry (MS) analysis, and data have also been complemented by molecular dynamics. Do the recent studies support our hypotheses? There now exist many studies on the co-translational folding of peptide nascent chains and of the structure of those peptides within the exit tunnel (some excellent recent reviews are, e.g., [42,43,44,45]). A considerable body of experimental evidence suggests that formation of α-helices within the ribosomal exit tunnel can occur for some proteins, in a manner that may be protein sequence, size and charge dependant [12,13,14,15,16,17,20,42,46,47,48]. In some other cases, there is evidence for compact, non-native structures that may represent nascent chains with discrete secondary structures that are not as coiled as full α-helices (e.g., [49]). In still other studies, there are suggestions that peptides remain in extended conformations (see, e.g., [19]). However, it must be cautioned that many studies carried out to date use (i) peptide chains that are unnaturally truncated, (ii) ribosomal expression that has been prematurely stalled, (iii) solution conditions that favour analytical methodology over bio-activity. Furthermore, in many studies, the structure of the peptides in the exit tunnel is not measured directly, but inferred by measuring the number of amino acids that need to be added to extend the nascent chain to a particular reporter group, leading to results that are open to debate.

Of course, the caveats mentioned at the end of the previous paragraph apply equally to work that is supportive and less supportive of our hypotheses. However, on balance, present experimental findings are supportive of our hypotheses—for example, there are few, if any, examples of β-sheet-like structures in the exit tunnel—and for more accurate structural data, more sophisticated experimental techniques should be developed that allow imaging of peptide chains in the exit tunnel in real time and in a cellular environment. Until now, there is a growing recognition of the key role that the ribosome plays in co-translational folding and that this may involve states that are either partly structured or else do not resemble the classic solution-state secondary structures [50], and evidence is beginning to emerge for helical starting conformations in peptides that ultimately will fold to β-structures [51].

From the possibility that the nascent chain is helical follows also a generic pathway for the early steps of folding. Namely, since β-sheets cannot form directly from the helix, in regions where the helix is not stable, the helix will evolve first into loops. I.e., the first step in folding is one in which the regions that are rich in G and/or P and/or N and/or D and/or S (the helix destabilizers) evolve into loops. Furthermore, β-sheets only arise with high probability if those regions happen to have V, and/or I, and/or F, and/or C, and/or W, in sufficient amounts. In this picture, the important factors in the folding process are the initial structure and the pathway. If/when we know both with sufficient accuracy we will be able to determine the native state from the amino acid sequence. I.e., we propose that determining the pathway is a more fruitful direction to follow than free energy minimization if you want to understand the protein folding process from a physical point of view.

It may be argued that single amino acid distributions are too limited and that longer sequences are needed for the definition of secondary structure. While this is true, it is nevertheless likely that such sequences will be composed of the structure-forming amino acids that have been identified in this study which, in turn, makes it unlikely that they will overturn the broad conclusions made here. Furthermore, as mentioned in the introduction, the variety of sequences with similar structures and the variety of structures with similar sequences makes the identification of such sequence-inducing structures very difficult. Thus, instead of trying to determine the sequence(s) capable of inducing a given secondary structure in the native state, we propose to look for pathway-defining sequences. More specifically, starting with the first step in folding mentioned in the previous paragraph, we should look for the regions where the nascent helix changes into loops. A previous study suggests that these regions should be bounded, on the N-side, by positively charged amino acids like K and H, and on the C-side by negatively charged amino acids like D [52].

Let us finish with two predictions which arise if the nascent chain is helical, and the pathway influences the native structure. One is that we expect that ribosome synthesis and chemical synthesis by a solid phase method [53,54] of the same proteins may lead to different structural outcomes. Namely, we predict that chemical synthesis will on average have a greater probability of leading to structures with more loops and sheets where ribosome synthesis of the same proteins leads to structures with a greater percentage of helices. A second prediction is that, if it is possible to make a hybrid ribosome in which the modern day decoding unit is coupled to the ancient synthesizing region that led to β-hairpins [37], then proteins that are largely composed of helices when synthesized by modern ribosomes will be mainly composed of sheets when synthesized by that hybrid ribosome. Both of these predictions challenge the thermodynamic hypothesis [2,3].

## Figures and Tables

**Figure 1 biomolecules-11-00357-f001:**
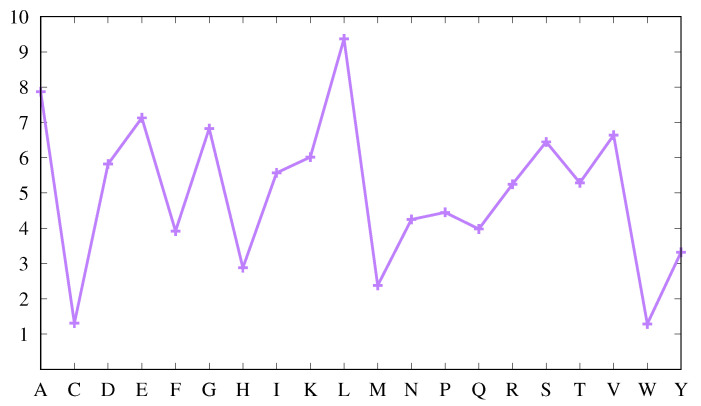
Average abundance, ${\bar{f}}_{a}\left(a\right)$ (cf. Equation (Equation 3)), of amino acid *a*, in the protein set used. The values are given in percentage of the total number of amino acids (see text).

**Figure 2 biomolecules-11-00357-f002:**
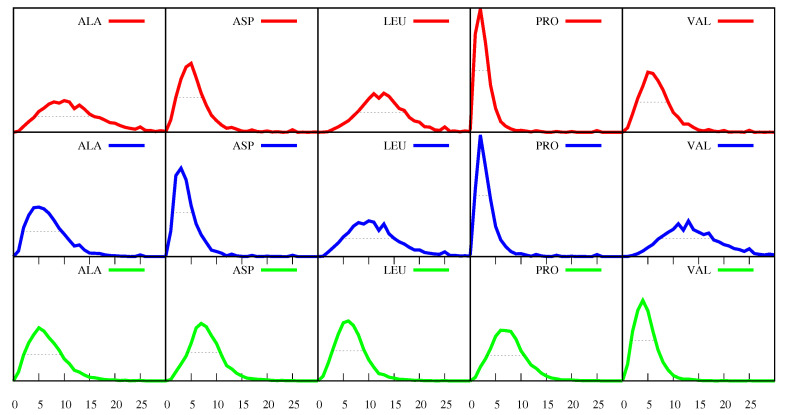
Distributions/histograms for a few of the frequencies f(a,s,p), where the amino acid *a* is specified at the top of each plot and where red is for s=α-helix, blue is for s=β-sheets and green is for s= loops. In this figure, the variable f(a,s,p) (the x-coordinate) runs from zero (which means that none of the amino acids *a* are found in *s*) to 30 (which means that 30% of the amino acids in the protein are *a*’s found in *s*). The y-coordinate is proportional to the number of proteins with a given value of f(a,s,p). The scale of the y-coordinate is the same for all plots and all distributions are normalised. The horizontal dotted lines are the FWHM of the distributions (see text).

**Figure 3 biomolecules-11-00357-f003:**
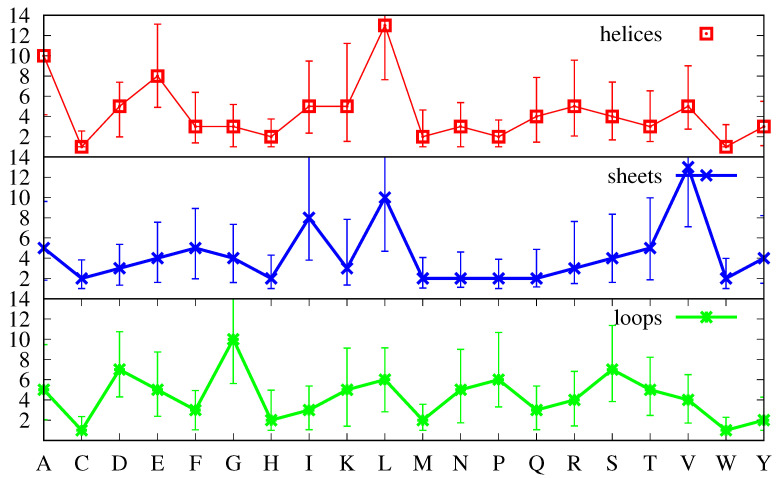
Average frequency of finding an amino acid *a* in α-helices (**top plot**), β-strands (**middle plot**) and loops (**bottom plot**). The values are given in percentage for each secondary structure, i.e., summing all the values in each line leads to 100. The amino acids are specified by their one letter codes.

**Figure 4 biomolecules-11-00357-f004:**
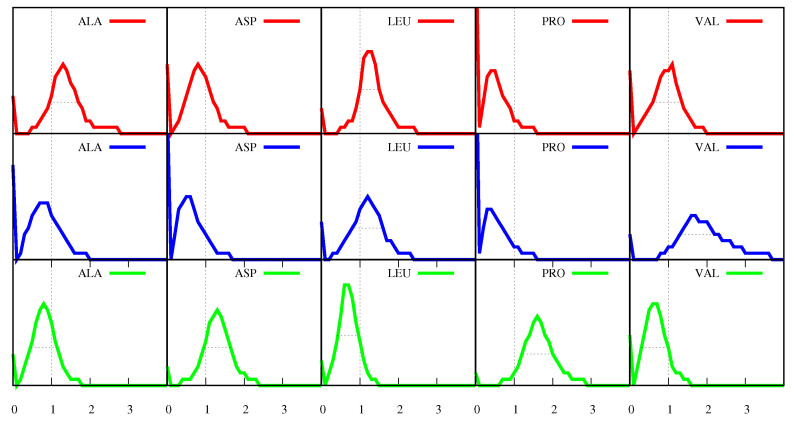
Distributions/histograms for a few of the ratios R(a,s,p) (cf. Equation (Equation 10)), where the amino acid *a* is specified at the top of each plot and where red is for s=α-helix, blue is for s=β-sheets and green is for s= loops. The variable R(a,s,p) (the x-coordinate) runs from zero to four, the scale of the ordinates is the same in all plots and all histograms are normalised. The vertical dotted line marks the value R(a,s,p)=1, when the actual number of amino acids *a* in secondary structure *s* is equal to what is expected in the absence of any correlation between *a* and *s*. The horizontal dotted lines are the FWHM of the distributions (see text).

**Figure 5 biomolecules-11-00357-f005:**
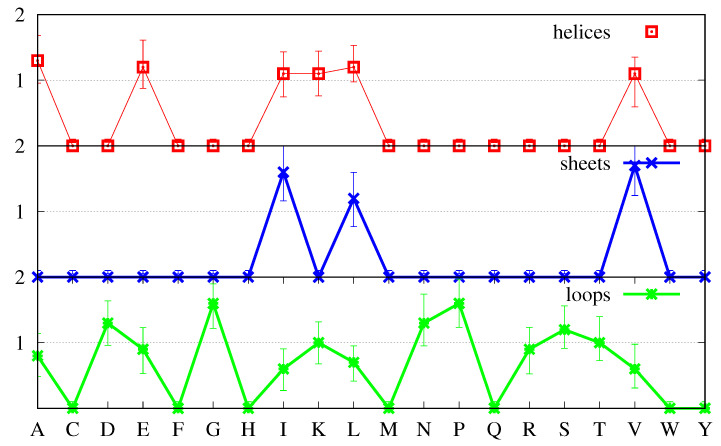
Most probable values for the ratios R(a,s,p) (cf. Equation (Equation 10)) for α-helices (**top plot**, red), β-sheets (**middle plot**, blue) and loops (**bottom plot**, green). The most probable value is taken from the full distributions for each ratio R(a,s,p) and the uncertainty around that value is given by the FWHM.

**Figure 6 biomolecules-11-00357-f006:**
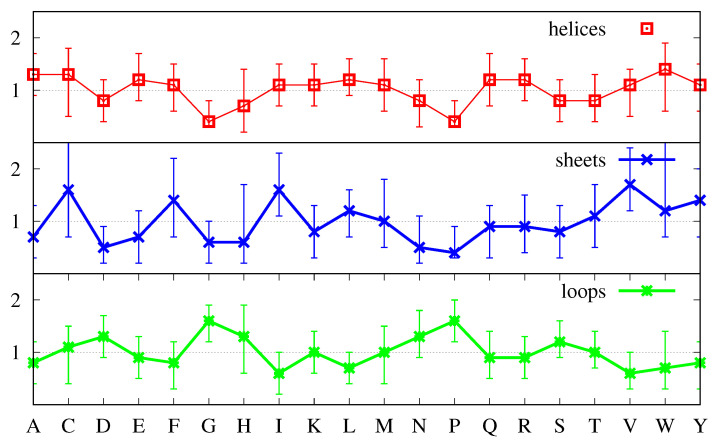
Most probable values for the ratios R(a,s,p) (cf. Equation (Equation 10)) for α-helices (**top plot**, red), β-sheets (**middle plot**, blue) and loops (**bottom plot**, green). The most probable value is taken from the middle peak in the distributions for each ratio R(a,s,p) and the uncertainty around that value is given by the FWHM of that middle peak.

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
