# Peer review of "Statistical Evidence for a Helical Nascent Chain"

_biomolecules, 2021, doi:10.3390/biom11030357_

Round 1

Reviewer 1 Report

The approach used in the manuscript is very simple and relies on frequencies of individual amino acid residues in protein sequences in order to estimate propensities for secondary structures and identify structure 'formers' and 'breakers'. As readily acknowledged, this works follows in the footsteps of Chou and Fasman and related (by now dated but still seminal) studies. While this could still represent an update worthy of publishing, I am not convinced the current manuscript meets that bar.

There are three major issues that need to be addressed:

i) How do the results compare with Chou and Fasman and related works? Is the inclusion of more sequences affecting the observed statistics or leading to new insights? Has the exclusion of lower quality structures (potentially leading to gaps in DSSP assigned states) affected the results? Are the multiple conformational states (such as those in NMR ensembles or multiple X-ray structures in different states) taken into account?

ii) Given the intriguing hypothesis of the dominance of helical structures in nascent protein chains emerging from ribosomes, the body of experimental work in this regard must be summarized in detail and specifically compared and contrasted with the predictions and speculations made in the manuscript. Also, what do the studies of re-folding with the aid of chaperons, as well as studies on conformational switches (between alpha and beta states in particular) complement the understanding of folding pathways and co-translational folding? What about membrane proteins, including beta barrels?

iii) Limiting the study to frequencies of individual amino acids in different secondary structures represents a major limitations that requires more critical analysis.

Authors try to side step this last issue by the following argument:

‘It may be argued that it is because single amino acids are not sufficient to determine the secondary structure and that amino acid sequences must be considered. This is certainly true, but this applies equally to β-sheets and loops.’

This however is not correct: helical structures are in general longer and imply periodicity and correlations within the secondary structure element being more important compared to beta strands; these correlations between multiple positions with several largely replaceable residues may result in the observed weaker propensities for alpha structures for individual amino acids.

Furthermore, local versus non-local contacts needed to stabilize the structure also play an important role, with beta structures stabilized by long range (in sequence) interactions, as indicated by the improved accuracy of predictions for beta strands when including the correct number of contacts or solvent accessibility for instance.

After all, the field of secondary structure prediction has gone a long way since the pioneering (and unsuccessful) efforts to predict such structure from amino acid frequencies and other simple statistics, accounting for all these effects through sophisticated machine learning approaches. 

Author Response

Response to Reviewer 1 Comments

General Comments about the changes made to the manuscript:

The revised manuscript includes two kinds of changes.

One kind consisted of re-organizing part of the text in the first half of section 3 in order to improve the presentation of our reasoning. This kind of change keeps much of the sentences of the previous version and mainly presents them in a different order. To mark these changes we have highlighted in blue the initial part of the text moved in the revised version and left in black the text that followed it in the previous version.

The second kind of changes are the additions or more extensive corrections to the text of the previous version, which are highlighted in red.

We hope this makes clear all the changes that have been made to the previous manuscript.

Answers to the specific points made by the Reviewer:

Point 1: The approach used in the manuscript is very simple and relies on frequencies of individual amino acid residues in protein sequences in order to estimate propensities for secondary structures and identify structure 'formers' and 'breakers'. As readily acknowledged, this works follows in the footsteps of Chou and Fasman and related (by now dated but still seminal) studies. While this could still represent an update worthy of publishing, I am not convinced the current manuscript meets that bar.

Response 1: We did not intend to update the analysis of Chou and Fasman who aimed at predicting secondary structures from the amino acid sequences. Instead, our aim is the reverse: it is to try to infer the initial structure (the structure of the nascent chain) from the amino acid distributions. We started with the single amino acid distributions because it seemed the proper place to start, and were in fact surprised to find that this most simple of all distributions already suggested a helical nascent chain.

In order to describe our aims better, the sentence "In this study we go back to the beginning and concentrate on single amino acid distributions." in the introduction of the previous version has been replaced by the following (in red in p.2, paragraph 1): "Another difficulty is that protein native structure may be one of the many kinetic traps into which the same polypeptide can find itself in, as shown in [23–25] and proposed in [26–30]. In this

case, reproducibility in always reaching the same native structure can be achieved if the initial structure and the pathway followed from it are always the same, as explained in detail in [28]. Thus, the purpose of our statistical analyses is to infer the structure of the nascent chain and of the generic features of the pathway."

Point 2: There are three major issues that need to be addressed:

  1. i) How do the results compare with Chou and Fasman and related works? Is the inclusion of more sequences affecting the observed statistics or leading to new insights? Has the exclusion of lower quality structures (potentially leading to gaps in DSSP assigned states) affected the results?

Response 2: We have added as supplemental information a comparison with the results of Chou and Fasman [1]. This shows that, in spite of the very short size of their protein set (only 15 proteins), many of the values of the ratios are similar.  In particular, the general trend that sheets and loops have more clearly identifiable structure-formers is already hinted at [1] (see figure 7 in file Suppl-information-Figures-1-11.pdf in supplemental information). To study further the effects of the size of the protein set, a calculation with a larger set, i.e. with all the 444520 proteins in the ss.txt file, was made which leads to results quite close to those presented in figures 1-6 in the main text, obtained with the non-redundant set of 13413 proteins, described in section 2. I.e., in all of the sets, structure-formers are much more clearly seen for sheets and loops, with the profile for alpha-helices displaying essentially clear structure-breakers (see the figures 1-8 in the file Suppl-information-Figures-1-11.pdf in supplemental information). Thus, in section 2 we have added in red the sentence: "In the file Suppl-information-Figures-1-11.pdf in supplemental information, figures 1-6 and 8 show the results that are obtained when the set of all the 444520 proteins, listed in the file list-of-444520-proteins.dat, is used."

Point 3: i) Cont.: Are the multiple conformational states (such as those in NMR ensembles or multiple X-ray structures in different states) taken into account?

Response 3: Multiple conformational states have not been taken explicitly into account but they are probably already represented by the inherent variety of protein structures included in the set and evident from the selected histograms of figures 1 and 4. Proteins are flexible molecules, they will certainly suffer structural fluctuations at biological temperatures, but here we rely on the correlations between their average structures and sequences.

Point 4: ii) Given the intriguing hypothesis of the dominance of helical structures in nascent protein chains emerging from ribosomes, the body of experimental work in this regard must be summarized in detail and specifically compared and contrasted with the predictions and speculations made in the manuscript. Also, what do the studies of re-folding with the aid of chaperons, as well as studies on conformational switches (between alpha and beta states in particular) complement the understanding of folding pathways and co-translational folding? What about membrane proteins, including beta barrels?

Response 4: According to the thermodynamic hypothesis, one function of chaperones is to accelerate folding, thereby preventing proteins from becoming locked in kinetic traps. Another possibility is that chaperones influence not just the speed but also the structural outcome of the refolding process. The fact that some of them, like GroEL, can refold proteins denatured in different ways, provides support for the latter possibility, but we can only speculate as to how they do it. For instance, one intriguing possibility is that GroEL may function to bring the misfolded proteins back to their initial stage, i.e., back to the helical form, but this remains conjecture for now. For integral membrane proteins, it is notable that their intra-membrane regions tend overwhelmingly to be made of helices. Since they are protected as soon as they emerge from the ribosome and until they are inserted in the membrane, and since they denature when taken out of the membrane, it may be that they are synthesized in the shape of helices to start with. Beta barrels are another special case, and our results provide a ready explanation for how they form, such that the beta strands arise from loops that have in turn arisen from unstable helices. In fact, this is also the case for proteins involved in protein-misfolding disorders, which predominantly involve an accumulation of beta sheet, often at the expense of helical sections of proteins. Thus, our results are fully in line with, and help to explain, diverse examples in normal and abnormal folding cascades.

We have added references 23-30 in support of our hypothesis and have added the following, in red, after the sentence "...in many cases, it is α-helical [12–17,20]." in section 4, paragraph 1, line 264:"It is also curious that the membrane regions of membrane proteins tend overwhelmingly to be made of helices. Since they are protected as soon as they emerge from the ribosome and until they are inserted in the membrane, and since they denature when taken out of the membrane, it does seem that they are synthesized in the shape of helices to start with. Furthermore, a specific mechanism for the formation of α-helices has been demonstrated for 5 different polypeptides in molecular dynamics simulations [30], namely, a forced rotation on the C-terminal, originating within the ribosome, lead to the formation of α-helices when the N-terminal outside is restrained [30]."

Point 5: iii) Limiting the study to frequencies of individual amino acids in different secondary structures represents a major limitations that requires more critical analysis.

Authors try to side step this last issue by the following argument:

‘It may be argued that it is because single amino acids are not sufficient to determine the secondary structure and that amino acid sequences must be considered. This is certainly true, but this applies equally to β-sheets and loops.’

This however is not correct: helical structures are in general longer and imply periodicity and correlations within the secondary structure element being more important compared to beta strands; these correlations between multiple positions with several largely replaceable residues may result in the observed weaker propensities for alpha structures for individual amino acids.

Furthermore, local versus non-local contacts needed to stabilize the structure also play an important role, with beta structures stabilized by long range (in sequence) interactions, as indicated by the improved accuracy of predictions for beta strands when including the correct number of contacts or solvent accessibility for instance.

After all, the field of secondary structure prediction has gone a long way since the pioneering (and unsuccessful) efforts to predict such structure from amino acid frequencies and other simple statistics, accounting for all these effects through sophisticated machine learning approaches.

Response 5: Figure 3 in the revised version shows all amino acids have a finite probability of appearing in all secondary structures. This means that all amino acids are tolerated by all the secondary structures and also that none is able, on its own, to either destroy, or lead to, the formation of a secondary structure. Thus, as the reviewer says, all secondary structures arise/survive because of correlations between at least a few of the amino acids that constitute them. These correlations may be related to periodicities (as the reviewer invokes for alpha-helices), or they may be due non-local contacts, as happens for beta-sheets, and of some other type for loops. Single amino acid distributions are not able to distinguish between these different types of correlations, but whatever their type, if the correlations exist, each of the amino acids involved should appear more often in the corresponding secondary structures when they favour/stabilize that secondary structures, and less often when they destabilize that secondary structure. More frequent amino acids than would arise in a random distribution are clearly found in sheets and loops but the absence of clear alpha-helix formers begs for an explanation. Ours is that alpha-helices do not need strong structure-formers because they come out of the ribosome already fully formed.

Reviewer 2 Report

I have read with interest this manuscript on the statistical occurrence of amino acid kinds in structures as beta-sheets and alpha-helices, and in loops of proteins. The analysis suggest some considerations that the Authors have collected in the final section. Some of them might sound speculative but overall the text is clear and the results are interesting.

Probably the manuscript deserves publication but the following points should be addressed first.

1) Figure 1 is quite confusing. In some columns there is the same amino acids, in other columns it can change with the row. Moreover, it does not contain the x-axis label. Why do the Authors not show all 20 cases well organized in columns?

2) I do not understand the logic of showing figure 2. Clearly amino acids that are less abundant in general will result less relevant in that kind of plot (it is the case, for instance, of tryptophan W). It is strange that this emerges only later in the text. Either the Authors make it clear from the beginning that this figure is just anticipating the real informative figures, or remove it entirely. In the statistical analysis, the comparison with the null model from eq.(11) could have been the start of the discussion, and eq.(10) would have been just enough, i.e. passages from figure 2 to equations (10,11) seem not essential.

3) The fraction defined in (4) is the average of this fraction over all proteins. This weights the contribution from long proteins less than an average over all amino acids of all proteins, i.e stacking everything in a sort of single long protein:
sum_p n_a(p) / sum_p n_p
I am wondering which of the two versions is more sensible. The Authors should motivate their choice.

4) I would find it surprising if the result shown in figure 3 about the abundance of amino acids is not known in the literature since decades. There should be proper citations to the relevant papers on this.

5) Are the fractions listed after eq(7) computed with eq(13)?

6) Is loop a structure? Talking about amino acid that are structure-formers for loops is at variance with the fact that loops should represent the more flexible and less structured part of proteins. Please clarify.

7) In the discussion about the factors making it difficult to form beta-sheets it should be mentioned that these are structures involving non-local contacts along the backbone. The local details cannot catch this structural difficulty, but it should be included in the speculations on why beta-sheets are less prevalent.

8) Irina Sorokina and Arcady Mushegian are among the people proposing that folding is not an equilibrium property but rather a dynamical transient cotranslational process. In relation to the final “prediction” of this work, one for example may also consider to quote https://www.biorxiv.org/content/10.1101/2020.09.01.277582v2

and references to their work therein.

Author Response

Response to Reviewer 2 Comments

General Comments about the changes made to the manuscript:

The revised manuscript includes two kinds of changes.

One kind consisted of re-organizing part of the text in the first half of section 3 in order to improve the presentation of our reasoning. This kind of change keeps much of the sentences of the previous version and mainly presents them in a different order. To mark these changes we have highlighted in blue the initial part of the text moved in the revised version and left in black the text that followed it in the previous version.

The second kind of changes are the additions or more extensive corrections to the text of the previous version, which are highlighted in red.

We hope this makes clear all the changes that have been made to the previous manuscript.

Answers to the specific points made by the Reviewer:

Point 1: I have read with interest this manuscript on the statistical occurrence of amino acid kinds in structures as beta-sheets and alpha-helices, and in loops of proteins. The analysis suggest some considerations that the Authors have collected in the final section. Some of them might sound speculative but overall the text is clear and the results are interesting.

Probably the manuscript deserves publication but the following points should be addressed first.

Response 1: In order to attend to comments 1) to 5) (points 2 to 6)  of reviewer 2, the results section was changed from its beginning in p.2, to essentially p.5, as explained above. As a result of these changes, figure 3 in the previous version is now figure 1, figure 1 in the previous version is now figure 2 and figure 2 in the previous version is now figure 3. As stated above, the parts of the revised version that differ from the previous because of small changes or because the order of presentation was inverted are in blue. In red are the texts that were added to the revised version. We hope that the report of our work now flows better.

Point 2: 1) Figure 1 is quite confusing. In some columns there is the same amino acids, in other columns it can change with the row. Moreover, it does not contain the x-axis label. Why do the Authors not show all 20 cases well organized in columns?

Response 2: We have attended to this comment by selecting the same amino acids in the three secondary structures. Also, we have identified the x-coordinate in the captions of the now figure 2 (as well as in figure 4, see aditions in red). On the other hand, a figure with all 20 cases organized in columns, as suggested by reviewer 2, will have a width of 3 units (one for each secondary structure) by a length of 20 units, i.e., its height will be more than 6 times longer than its width, which is awkward. The aim of showing only a few selected histograms was to show the type of curves that arise and to make clear where the most probable values and FWHM plotted in figure 3 (previous figure 2) came from. For the sake of completeness, all the histograms, for the three secondary structures are displayed in figures 9-11 of the file Suppl-information-Figures-1-11.pdf in supplementary information.

Point 3: 2) I do not understand the logic of showing figure 2. Clearly amino acids that are less abundant in general will result less relevant in that kind of plot (it is the case, for instance, of tryptophan W). It is strange that this emerges only later in the text. Either the Authors make it clear from the beginning that this figure is just anticipating the real informative figures, or remove it entirely. In the statistical analysis, the comparison with the null model from eq.(11) could have been the start of the discussion, and eq.(10) would have been just enough, i.e. passages from figure 2 to equations (10,11) seem not essential.

Response 3: We agree that some of our main points could be presented better and for that reason we have made changes to the first part of section 3, as mentioned above. With these changes, we hope that in this revised version it is clear that comparing figure 3 (which was previously figure 2) with figure 1 (which was previously figure 3) one can already surmise the existence of amino acid bias because not only are the amino acid distributions different for each secondary structure, but also they are different from the average abundance in figure 1. On the other hand, a further point is that the values in figure 3 are still not the best estimates of the bias for the reasons given from the last line in p.5 to Eq.(8) in p.7.

Point 4: 3) The fraction defined in (4) is the average of this fraction over all proteins. This weights the contribution from long proteins less than an average over all amino acids of all proteins, i.e stacking everything in a sort of single long protein:

sum_p n_a(p) / sum_p n_p

I am wondering which of the two versions is more sensible. The Authors should motivate their choice.

Response 4: To motivate our choice, we have taken out the following sentence in p.2, paragraph 2, of the previous version: "While in some statistical studies the sequences of many proteins are all lumped together [22,23], the sequence-structure variety mentioned in the previous paragraph can best be gauged when each protein is considered separately" and have added the following sentence (in red in p.3, after Eq.(3)): "In many studies (see e.g. [1,35,36]), all proteins are mixed together into one single enormous protein and the statistics are calculated for this single protein. However, such a single protein does not exist and it is impossible to know what its structure would be. This is one reason why, in Eq.(3) as well as in all averages over the protein set that are mentioned below, we consider the statistics for each protein separately, and then average over all the proteins in the set. A second reason is that, as found in [36] and as illustrated in figures 2 and 4 below, those separate statistics can vary significantly from protein to protein." Furthermore, a specific mechanism for the formation of α-helices has been demonstrated for 5 different polypeptides in molecular dynamics simulations [30], namely, a forced rotation on the C-terminal, originating within the ribosome, lead to the formation of α-helices when the N-terminal outside is restrained [30]."

Point 5: 4) I would find it surprising if the result shown in figure 3 about the abundance of amino acids is not known in the literature since decades. There should be proper citations to the relevant papers on this. This however is not correct: helical structures are in general longer and imply periodicity and correlations within the secondary structure element being more important compared to beta strands; these correlations between multiple positions with several largely replaceable residues may result in the observed weaker propensities for alpha structures for individual amino acids.

Response 5: We have added the following sentence in p.3, paragraph 4, in red:"However, it is also known that the average amino acid abundance depends on protein size [36]. The protein set used here includes proteins with sizes from 20 to 1859 amino acids, with a broad peak at 157. Comparing with values obtained in [36] for proteins with an average size of 200 amino acids, the abundances are similar. Furthermore, using the larger data set mentioned in the section 2, the results are virtually indistinguishable (compare figure 1 above with figure 1 in file Suppl-information-Figures-1-11.pdf of supplemental information). This validates our protein set from the point of view of average amino acid composition."

Point 6: 5) Are the fractions listed after eq(7) computed with eq(13)?

Response 6: We hope that these doubts have been explained by the improved presentation and the new order of the equations in the present version.

Point 7: 6) Is loop a structure? Talking about amino acid that are structure-formers for loops is at variance with the fact that loops should represent the more flexible and less structured part of proteins. Please clarify.

Response 7: As explained in p.4 paragraph 2, the designation "loops" in this study includes beta-bridges, turns, bends, as well as random coil. The first three are indeed considered as proper secondary structures, but random coil less so. The point here was to distinguish between alpha-helices and beta-sheets and the less structured regions that connect the first two. Of course, we agree with the reviewer that random coil is essentially the absence of structure. Still, even in CASP14, protein structures are evaluated for fixed positions and conformations of such "loops". From our point of view there is one essential initial structure, the helix, and the other two, sheets and loops, arise through instabilities of the former. In order to make this point clearer we have added the following sentence in p.4 after Eq.(4):"In this way, we separate the helices and sheets from the less structured regions that connect them."

Point 8: 7) In the discussion about the factors making it difficult to form beta-sheets it should be mentioned that these are structures involving non-local contacts along the backbone. The local details cannot catch this structural difficulty, but it should be included in the speculations on why beta-sheets are less prevalent.

Response 8: In the first part of the discussion we try to explain the observed secondary structures and their frequency from the point of view of the thermodynamic hypothesis. According to the thermodynamic hypothesis, beta-sheets, just as any other structure, form because they are the global free energy minimum. From this point of view, the fact that they are stabilized by non-local contacts may increase the time it takes for beta-sheets to form but it cannot explain why they are less prevalent. Equating a longer time for formation with a decreased prevalence already means taking into account a dependence of the final structure on the pathway, which is indeed where we end up  (please see paragraph 3 in p.13).

In this study we have only used single amino acid distributions, but this does not mean that we have only considered local details. Indeed, figure 3 in this revised version shows that all amino acids have a finite probability of appearing in all secondary structures. This means that all amino acids are tolerated by all the secondary structures and also that none is able, on its own, to either destroy, or lead to, the formation of a secondary structure. Thus, all secondary structures arise/survive because of correlations between at least a few of the amino acids that constitute them. These correlations may be may be due to local, or to non-local contacts (as happens for beta-sheets). Single amino acid distributions are not able to distinguish between these different types of correlations, but whatever their type, if the correlations exist, each of the amino acids involved should appear more often in the corresponding secondary structures when they favour/stabilize that secondary structures, and less often when they destabilize that secondary structure. In spite of the need for non-local contacts in beta-sheets, their distributions do clearly show the existence of more frequent amino acids than would arise in a random distribution. These are most probably those involved in the non-local contacts.

Point 9: 8) Irina Sorokina and Arcady Mushegian are among the people proposing that folding is not an equilibrium property but rather a dynamical transient cotranslational process. In relation to the final “prediction” of this work, one for example may also consider to quote https://www.biorxiv.org/content/10.1101/2020.09.01.277582v2

and references to their work therein.

Response 9: Thank you for bringing those authors to our attention. They share our hypothesis that protein folding in vivo is a kinetic process and have done molecular dynamics simulations on how alpha-helices can form just outside the ribosome, while the nascent chain is still being synthesized. We have added references to their work in this revised version (references 29 and 30) and also the following sentence in p. 13, paragraph 1: "The proposal in [27] is that the helix forms while still inside the ribosome but another possibility is that it forms right outside as a result of constraining the nascent chain while rotating it inside the ribosome, as found in molecular dynamics simulations [30]."

Round 2

Reviewer 1 Report

While I can appreciate the effort to improve clarity, the revised version still fails to address the issue of experimental support for the main hypothesis of the manuscript, i.e., the claim that nearly all nascent chains adopt helical conformations and subsequently unfold/refold into other (beta) structures. Experimental studies on the structure of nascent chains must be carefully reviewed and the weight of evidence in support vs. against the claim considered. Even one of the recent papers cited (but not discussed/debunked) by the authors, namely ref. 19, Nat Comm 2018, is titled ' Transmembrane but not soluble helices fold inside the ribosome tunnel.' which appears to directly contradict the central claim of the manuscript. Other experimental studies suggest extended conformation as most likely, while only short helical segments are observed in co-translational folding pathways (see e.g. Bhushan, S. et al. alpha-Helical nascent polypeptide chains visualized within distinct regions of the ribosomal exit tunnel. Nat. Struct. Mol. Biol. 17, 313–317 (2010)). In other words, while there is evidence of helical structures being formed inside the ribosomal tunnel, the leap to a conclusion that nearly all peptides go through helical intermediates requires that readers are presented the whole body of evidence for and against it.

Author Response

There is already a very considerable number of studies published on the structure of the nascent chain. We have added 14 new references to those already quoted before, but we do not have the space available to consider each in detail. However, as was asked, we have endeavoured to provided a more considered review of the current literature in the discussion section, some of which is supportive of our conclusions and other not so much. More specifically, we have added in red, in p.13 of this second revised version, the following text: "Indeed, some recent discussions of ribosome evolution suggest that the exit tunnel has evolved to favour formation of helical segments [37]. The dimensions of
this tunnel place restrictions on the secondary structural elements that can form in nascent chains during translation [38], particularly in the first 50 Å or so from the peptidyl transfer complex where the diameter is
tightly constrained [39–41]. In recent years a range of sophisticated biochemical and biophysical tools have been developed to study the structure of nascent chains complexed inside the exit tunnel, largely driven by the availability of cell-free translation preparations and by our ability to pause translation in controlled manners. Analytical tools that have been applied include cryo-electron microscopy, protein labelling and crosslinking approaches, nuclear nagnetic resonance (NMR) and mass spectrometry (MS) analysis, and data has also been complemented by molecular dynamics. Do the recent studies support our hypotheses? There now exist many studies on the co-translational folding of peptide nascent chains and of the structure of those peptides within the exit
tunnel (some excellent recent reviews are e.g. [42–45]). There is a considerable body of experimental evidence to suggest that formation of α-helices within the ribosomal exit tunnel can occur for some proteins, in a manner that may be protein sequence, size and charge dependant [12–17,20,46–49]. In some other cases, there is more evidence for compact, non-native structures that may represent nascent chains formed into discrete secondary structures that are not as coiled as full α-helices (e.g. [50]). In still other studies, there are suggestions that peptides remain in extended conformations (see e.g. [19]). However, it must be cautioned that many studies carried out to date use (i) peptide chains that are unnaturally truncated, (ii) ribosomal expression that has been prematurely stalled, (iii) solution conditions that favour analytical methodology over bio-activity. Furthermore, in many studies the structure of the peptides in the exit tunnel is not measured directly, but inferred by measuring the number of amino acids that need to be added to extend the nascent chain to a particular reporter group, leading to
results that are open to debate. The caveats apply equally to work that is supportive and less supportive of our hypotheses. However, on balance, experimental findings are supportive of our hypotheses – for example, there are few, if any, examples of β-sheet-like structures in the exit tunnel – but more sophisticated experimental techniques should be developed to allow imaging of peptide chains in the exit tunnel in real time and in a
cellular environment. Nevertheless, there is a growing recognition of the key role that the ribosome plays in co-translational folding and that this may involve states that are either partly structured or else do not
resemble the classic solution-state secondary structures [51], and evidence is beginning to emerge for helical starting conformations in peptides that ultimately will fold to β-structures [52]."